# Perioperative Analgesia and Patients’ Satisfaction in Spinal Anesthesia for Cesarean Section: Fentanyl Versus Morphine

**DOI:** 10.3390/jcm12196346

**Published:** 2023-10-03

**Authors:** Mihai O. Botea, Diana Lungeanu, Alina Petrica, Mircea I. Sandor, Anca C. Huniadi, Claudiu Barsac, Adina M. Marza, Ramona C. Moisa, Laura Maghiar, Raluca M. Botea, Codruta I. Macovei, Erika Bimbo-Szuhai

**Affiliations:** 1Department of Surgery, Faculty of Medicine and Pharmacy, University of Oradea, 410087 Oradea, Romania; drmob78@yahoo.com (M.O.B.); drims75@yahoo.com (M.I.S.);; 2Pelican Clinic, Medicover Hospital, 4104869 Oradea, Romania; 3Center for Modeling Biological Systems and Data Analysis, “Victor Babes” University of Medicine and Pharmacy, 300041 Timisoara, Romania; dlungeanu@umft.ro; 4Department of Functional Sciences, “Victor Babes” University of Medicine and Pharmacy, 300041 Timisoara, Romania; 5Department of Surgery, “Victor Babes” University of Medicine and Pharmacy, 300041 Timisoara, Romania; claudiu_barsac@yahoo.com (C.B.); marza.adina@umft.ro (A.M.M.); 6Emergency Department, “Pius Brinzeu” Emergency Clinical County Hospital, 300736 Timisoara, Romania; 7Clinic of Anaesthesia and Intensive Care, “Pius Brinzeu” Emergency Clinical County Hospital, 300736 Timisoara, Romania; 8Emergency Department, Emergency Clinical Municipal Hospital, 300079 Timisoara, Romania; 9Oradea County Clinical Emergency Hospital, 410169 Oradea, Romania

**Keywords:** cesarean section, spinal anesthesia, analgesia, pain scores, bupivacaine, fentanyl, morphine

## Abstract

Perioperative analgesia for cesarean section aims to ensure the mother’s comfort, facilitate a smooth surgical experience, and promote a successful recovery. One-hundred-ninety patients were enrolled in a randomized double-blind study designed to assess the quality of perioperative analgesia, level of satisfaction, and incidence of adverse reactions in elective cesarean section under spinal anesthesia when fentanyl or morphine was added to bupivacaine. Two treatment groups comprising 173 subjects were compared in the per-protocol analysis: F (fentanyl, standard dose 25 μg) and M (morphine, standard dose 100 μg). Numerical pain scores were recorded perioperatively for 72 h (both at rest and on mobilization), with overall postoperative satisfaction and analgesic-related side effects. The patients in the morphine group had significantly better pain management (Mann–Whitney U test, *p* < 0.001) and higher level of satisfaction (Mann–Whitney U test, *p* < 0.001). The latter was related to the greater need for rescue medication in the fentanyl group (OR = 4.396; *p* = 0.019). On the other hand, fentanyl had significantly fewer non-life-threatening side effects, such as high-intensity pruritus (Mann–Whitney U test, *p* < 0.001), nausea (OR = 0.324; *p* = 0.019), vomiting and dizziness upon first mobilization (OR = 0.256; *p* < 0.001). It remains for future clinical trials to help establish doses that will tilt the scale to one side or the other.

## 1. Introduction

Cesarean section is a common surgical procedure for delivering a baby and can result in significant pain for the parturient. Postpartum pain is an important and ongoing concern for women and an important factor of maternal satisfaction [1,2]. Acute postpartum pain is an independent risk factor for the subsequent development of chronic postpartum pain [3,4] and is also a predictor for depression [5]. Postpartum pain control is essential for an optimal functional recovery and is likely to influence other recovery aspects, like physical function, mother‘s comfort, emotional and social wellbeing, sleep quality, optimal breastfeeding, and maternal–neonatal bonding [6].

Pain is a complex and multidimensional experience. Furthermore, pain severity scores do not consider influences by other factors, such as fear, depression, anxiety, nausea, vomiting, or functional limitations [6]. This is the reason we should consider overall patient’s satisfaction aside with the pain scores and the incidence of anesthesia adverse effects to estimate the perioperative patient‘s comfort.

There are two approaches to anesthesia for cesarean section:
1.Regional neuraxial anesthesia—includes techniques such as spinal anesthesia, epidural anesthesia, or combined spinal–epidural anesthesia. These techniques provide effective pain relief during and after the surgery, spinal anesthesia being the technique of choice.2.General anesthesia—is only used if the patient’s medical condition requires it, if it is the mother‘s choice, or in the case of an emergency with no time to perform a regional neuraxial block.

Most anesthetists have a multimodal analgesia approach for postoperative pain control, namely a combination of regional techniques (neuraxial or peripheral nerve blocks) and systemic analgesia (such as regular systemic analgesics or patient-controlled analgesia), sometimes combined with non-pharmacological methods, aiming to provide comprehensive pain management. So far, systemic analgesics such as opioids are the mainstay of analgesia, especially in the early postoperative hours, but the current trends aim at decreasing reliance on this class of drugs due to the high rates of adverse reactions. The choice regarding the anesthesia technique and perioperative analgesia strategy depends on several factors: patient’s medical history, type of surgery, and patient’s preference. The anesthetist and obstetrician must work together to determine the best approach for each individual patient and establish side effects management and rescue analgesia plans.

Different regimens of multimodal analgesia have been tested in clinical trials. An intrathecal combination of a local anesthetic and an opioid is frequently used. Fentanyl (doses of 10–25 μg) and morphine (doses of 0.1–0.2 mg) are two regularly used opioids for obstetrical pain relief, each with distinct advantages regarding the onset and duration of their effects (9). Intrathecal fentanyl used as an adjunct has many advantages: it improves the quality of spinal anesthesia by acting quickly and therefore decreasing the time to establish the block [7], and it has a bupivacaine-sparing effect with fewer episodes of hypotension during surgery [8,9]. The onset of morphine’s effects is slower, but it lasts for a longer period, producing an excellent analgesia after surgery [10]. However, there is still no agreement upon the best choice of opioid and dosage [10,11], and their comparative effectiveness remains unknown [12].

The objectives of this study were (a) assessing the perioperative analgesia and the degree of patient‘s satisfaction when comparing morphine 100 μg with fentanyl 25 μg used as an adjunct to bupivacaine in spinal anesthesia for cesarean section; (b) assessing the incidence and the degree of side effects, like nausea, vomiting, dizziness, pruritus, sedation, and respiratory depression.

## 2. Materials and Methods

### 2.1. Study Design and Participants

We conducted a prospective double-blind, randomized trial, from January to December 2022 at Pelican Clinic Hospital from Oradea (Romania). The study was conducted in accordance with the Declaration of Helsinki, and the protocol was approved by the Ethics Committees of Pelican Hospital from Oradea (no. 2672/28.12.2021). Full-term parturient patients scheduled for cesarean section under spinal anesthesia were randomly distributed in two groups: group F (fentanyl) was administered hyperbaric bupivacaine 0.5% with a standard dose of 25 μg fentanyl; group M (morphine) received hyperbaric bupivacaine 0.5% with a standard dose of 100 μg morphine. The dose of hyperbaric bupivacaine 0.5% to be injected was based on the patient’s height (doses range between 7.5 mg and 11 mg).

Figure 1 shows the study design. The explicit inclusion and exclusion criteria are included in this flowchart. This study is registered on ClinicalTrials.gov under the number NCT05533229.

The inclusion criteria were ASA I-II patients; age 18–40 years; no medical history; unknown allergies to the used medication; no history of chronic pain or regular use of analgesics; no history of anxiety or depression; body weight ≥ 50 kg; elective cesarean section indication with a pregnancy equal to or above 37 weeks; alive and single fetus.

The exclusion criteria were pregnant women with psychiatric disorders; history of drug addiction; diagnosis of acute or chronic fetal distress; contraindication to spinal anesthesia; patient refusal; preeclamptic patients; patients who developed allergic reaction after enrolling in the study; refusal of analgesia; need of surgical re-intervention in the next 72 h after cesarean section; conversion from a natural delivery with/without an epidural anesthesia in place; previous administration of opioids and/or other central nervous system depressants.

During the pre-anesthetic consultation, all patients received information about the aims of the study, the choice of anesthesia, and details about the pain management and assessment using the numerical pain scale (NPS). Written consent was obtained after being assessed for inclusion criteria. Two anesthetists were needed to ensure the double-blind compliance: one randomly assigned the patient and prepared the intrathecal solution for the spinal anesthesia; another one performed the spinal anesthesia (blinded to the used solution). All drugs were from the same manufacturer.

### 2.2. Variables under Examination

Patients’ demographics data comprised height, term weight, and age. In addition, drugs and full medical history, previous allergies, and other relevant medical data were documented (gesta/para status, preoperative uterine contractions, time from anesthesia to incision, time to sensory block). The pain assessment was carried out through an interview conducted by a member of the medical staff, namely a senior anesthesia trainee. The assessment and follow-up staff were independent of the team who provided anesthesia. Pain was assessed on an 11-point numeric rating scale (NRS) from 0 (no pain) to 10 (worst imaginable pain). Participants’ subjective pain intensity was recorded at the baseline during the surgical intervention (PS0), at the end of the cesarean section (PS1), and at 4, 6, 12, 24, 48, and 72 h after PS1. Postoperative pain scores were assessed at rest and on mobilizing.

In case of required postoperative rescue medication for pain, imputation was subsequently applied: last value was carried forward within a four-hour time window. To comprehensively compare the postoperative pain in the study groups, the area under the pain scores over 72 h (AUPS72h) was employed:(1)$${AUPS}_{72h}=\sum_{t1}^{72h}\left({PSD}_{t_{i}}+{PSD}_{t_{i-1}}\right)/2*\left(t_{i}-t_{i-1}\right)$$
where $t_{i}$ is the time of measurement (with i = 0, 1, 4 h, 6 h, 12 h, 48 h, 72 h); and pain scores’ differences are defined as ${PSD}_{t_{i}}={PS}_{t_{i}}-{PS}_{baseline}$.

Adverse effects associated with anesthesia were recorded: presence and intensity of pruritus, nausea, vomiting, sedation, respiratory depression, dizziness (either constant or on first mobilization). Level of patients’ overall satisfaction was recorded on a 5-point Likert-type scale.

### 2.3. Sample Size

The sample size of the two trial groups was determined for a 1:1 ratio, considering two types of outcomes: (a) the pain score on a 0 to 10 scale; (b) the proportion of symptoms associated with anesthesia. The R packages “MKpower” v. 0.5 (applying Monte-Carlo simulation to determine empirical power) and “webPower” v. 0.6 applying the J. Cohen theory were employed [13]. The simulation was conducted for the following data: mean pain scores of 1.5 and 2.5, respectively, with equal standard deviation of 2; alpha = 0.05; power ≥ 0.8. For proportions, a small to medium effect size h = 0.45 was considered; alpha = 0.05; power = 0.8. For both methods, two-sided tests were considered. The resulting sample size was 79 subjects in each group. A 20% estimated dropout adjustment was applied [14,15], so 95 patients were initially randomized in each group.

The actual dropout differed in the two groups, so they ended up having different sizes, as shown in Figure 1. To reveal the difference between the two anesthesia approaches, per protocol analysis (i.e., on-treatment) was decided from the beginning. Overall, there was no decreasing effect on the statistical power.

### 2.4. Medical Procedures’ Protocol

Preoperative venous access was obtained with an 18 G cannula, and 500 mL of Ringer lactate solution was infused prior to anesthesia. Patients were fasted and given pantoprazol 40 mg and metoclopramide 10 mg (intravenously as pre-anesthetic medication). The spinal anesthesia was performed under an aseptic technique with the patient in a sitting position, at the level of L3–L4 interspace, using a midline approach, with a 27 G Whitacre needle and an introducer. The anesthetic mixture was manually injected at a rate of 1 mL.15 s^−1^, with a barbotage effect. Local anesthetic infiltration was performed with 2 mL of 1% lidocaine. After the block, the patient was placed in a supine position with a roll wedge placed under the right hip to displace the uterus to the left until fetal extraction.

Warm Ringer lactate (10–20 mL kg^−1^ hr^−1^) was given to optimize the volemic status. All patients had anti-embolic socks on lower limbs. Surgery started when sensory block reached the T4 dermatome level, as assessed by the cold test.

Non-invasive blood pressure (BP) was measured every minute until the baby was delivered and every 3–5 min afterwards. Any decrease in systolic blood pressure below the baseline level resulted in intravenous ephedrine doses of 5–15 mg, repeated every minute until the systolic blood pressure was optimally restored. At delivery, oxytocin 5 I.U. was given intravenously (normal practice when the study was conducted) and afterwards as required.

Our approach to prevent the emetic syndrome involved an initial dose of 1 mg granisetron given intraoperatively and a single dose of 4 mg dexamethasone. A second dose of granisetron was administered 12 h apart. Afterwards, anti-emetic medication was given if required.

Heart rate, blood pressure, and oxygen saturation measured by pulse oximetry (SpO2), requirements for supplemental analgesics and the need for conversion to general anesthesia, were recorded intraoperatively. In addition to these parameters, the level of sensory block was assessed by the cold test, and the maximum motor block score was recorded using the 4-point Bromage scale until complete motor block recovery.

Postoperative pain control strategy was based on regular paracetamol 1 g every 6 h and ibuprofen 400 mg every 8 h. Doses of 50 to 100 mg of intravenous tramadol were available if required, to a maximum dose of 400 mg/day.

### 2.5. Data Analysis

Descriptive statistics included the observed frequency counts (percentage) for categorical variables and mean ± standard deviation for numerical variables, irrespective of their distribution. Normality was tested with the Kolmogorov–Smirnov test. For comparing means in normally distributed values, the *t*-test for independent samples was applied, with Levene’s test for equality of variances. For comparing distribution of non-normally distributed numerical values, the non-parametric Mann–Whitney U statistical test was applied and median (interquartile range) with Tukey’s hinges was additionally provided as a descriptive statistic.

The chi-squared statistical test (either asymptotic, Fisher’s exact test, or Monte-Carlo simulation with 10,000 samples) was applied to check the statistical significance of the association between the categorical variables. The odds ratio (OR) values were calculated for the symptoms associated with the two anesthetics, such as nausea and dizziness.

The statistical analysis was conducted at a 95% level of confidence and a 5% level of statistical significance. All reported probability values were two-tailed. Statistical analysis was performed with the statistical software IBM SPSS v. 20 and open-source R v. 4.0.5 packages.

## 3. Results

The study enrolled 190 full-term parturient patients scheduled for cesarean section under spinal anesthesia. After the randomization, seventeen patients were excluded due to various reasons: participation declining after enrollment (six patients); developing allergic reactions (four patients); pre-eclampsia development (four patients); and surgical re-intervention (three patients). Therefore, the final analysis was conducted on 173 patients: 94 in M group, and 79 in F group.

Table 1 presents the demographic and clinical characteristics of the patients. Statistically significant differences were observed only in their smoking habits (*p* = 0.039), with a higher percentage of smokers in the M group (18.1% compared to 8.9%).

The mean time from anesthesia to incision for all patients was 6.50 min. The full data regarding the time elapsed from anesthesia to sensory block and surgical incision are shown in Table 2. There are no significant differences in the time intervals related to anesthesia and sensory block between the patients of the two study groups.

The results for perioperative pain are presented in Table 3. There were significant differences in pain intensity scores between the M and F groups regarding the comprehensive metrics of AUPS72h (both at rest and on mobilization), and at all postoperative recording times, with evidence of better and long-lasting analgesic effects in M patients (both at rest and on mobilization).

Figure 2 shows the pain scores’ evolution in time for the two groups (a and b, at rest and on mobilization, respectively). The pain scores during the surgery (PS0) and immediately after (PS1) are depicted in both diagrams as references. PS0 is the baseline value in Equation (1), used to calculate AUPS72h.

Figure 3a,b illustrate the AUPS72h scores as comprehensive measures of the overall pain management in the two study groups. The pain management was arguably better in M group across all the metrics.

Table 4 synthesizes the occurrence of adverse effects and experiences, such as pruritus, nausea, vomiting, and dizziness. Moderate to severe pruritus was reported more often in M group (14.9% compared to 8.1%), while vomiting was encountered only in this group. Patients in F group experienced nausea in a significantly smaller proportion compared to M group (20.2% versus 7.6%), with OR = 0.324 and 95%CI (0.123; 0.858). Neither sedation nor respiratory depression was reported in any of the groups. Patients in F group also experienced dizziness in a smaller proportion: none of them reported constant dizziness (compared to three individuals who reported the condition in M group); nineteen (24.1%) subjects in F group reported dizziness upon first mobilization compared to fifty-one (55.3%) in M group; the odds of dizziness were significantly smaller in F group, with OR = 0.256 and 95%CI (0.133; 0.493).

Table 5 shows the higher proportion of required rescue analgesic medication in F group, namely ten (12.7%) patients compared to three (3.2%) in M group. The odds of postoperative additional analgesics were significantly higher in F group, with OR = 4.396 and 95%CI (1.165; 16.582).

Contrary to the findings regarding higher incidence of adverse side effects in M group, these patients had better perception of analgesia effectiveness and higher satisfaction levels compared to F group (Table 6). The differences in both criteria were statistically significant (*p* < 0.001) in favor of M group. The individual characteristics of the 13 patients who required rescue analgesia are presented in Appendix A.

Figure 4 illustrates this balance in the advantages and disadvantages of postoperative adverse events in the two study groups: the OR values for the symptoms associated with anesthesia (favoring F) on the one hand and the need for additional analgesics (favoring M) on the other hand.

## 4. Discussion

Spinal anesthesia is a common technique in the field of obstetric anesthesia as it is accessible, safe, and easy to perform. In addition to an optimal surgical anesthesia, by using a local anesthetic agent, an adequate relaxation of the abdominal wall muscles can be induced. It is well known that mixing opioids with bupivacaine will allow a decrease in the local anesthetic dose with similar efficiency on pain control and even better hemodynamic stability, being also a simple and effective way to manage postoperative pain in cesarean section patients [16].

In our study, 77.2% of the fentanyl group patients reported effective pain control, while all 94 patients (100%) from the morphine group reported effective analgesia and consistently rated their satisfaction as maximum. Only three patients from the morphine group needed rescue medication. In contrast, 10 patients from the fentanyl group needed additional analgesics. Our study results showed the analgesia to be more effective in the morphine group during the postoperative hours. Similar results were reported by Karaman et al. [10] and El Aish et al. [17]. The latter study also reported greater time elapsing before the need for additional analgesics in the case of morphine. The odds of such rescue medication reported in the case of fentanyl were OR = 2.59, 95% CI (2.03; 3.31). Our results led to higher odds but also a considerably larger confidence interval, namely OR = 4.396 with 95%CI (1.165; 16.582).

According to some researchers, intrathecal morphine reduces intraoperative discomfort [18,19], whereas other authors concluded that it only starts to work postoperatively [20]. Intraoperative pain was reported in 18% to 29% of cases after administration of spinal morphine at a dose of 0.1 to 0.2 mg [20,21,22,23]. Fournier and Baraka reported the onset of morphine action to be 30 to 60 min after spinal administration [24,25]. In more recent research, Wojciech et al. concluded that morphine did not have any intraoperative analgesic effect and that 25% of women needed additional intra-operative analgesia [26]. Other studies pointed out a similar result, raising concerns on the decrease in intraoperative analgesia [10,26]. A combination of fentanyl with a local anesthetic in obstetrics anesthesia was reported as equally effective in doses much smaller than those used in our study [10,22,26,27]. Research in animal models found an escalating influence of progesterone as an analgesic effect of lipophilic spinal opioids during pregnancy [10,18]. The results of our study contribute to the evidence of morphine being as potent as fentanyl for assuring an efficient intra-operative surgical anesthesia.

Concerning the duration of effective analgesia, previous studies pointed out that the use of fentanyl in combination with a local anesthetic had about a 12 h effect, whereas the average duration of effective analgesia was 18 to 22 h in studies when morphine 100 μg was used [10,18].

Regarding opioids’ side effects, previous research [11,18,19,20] indicated that the inclusion of morphine in a local anesthetic agent would lead to a higher likelihood of pruritus, typically ranging from 40% to 63%. In our study, a moderate to severe pruritus was encountered in only 8.1% of cases. The minor pruritus was bearable most of the time, with no clinical relevance.

Mild nausea and drowsiness were more frequently reported by participants receiving morphine compared to fentanyl, at 6 h and 12 h, respectively [17]. Our patients in the morphine group developed nausea in 20.2% of cases (compared to 7.6% in the fentanyl group) and vomiting in 12.8% of cases (compared to none in the fentanyl group). This overall lower incidence was probably due to the protective role of the granisetron and the dexamethasone as single doses routinely administered during the surgical intervention.

Other studies also reported problems with side effects, such as nausea and vomiting [28,29]. Researchers suggested that approximately 40% of patients might experience nausea and 15% to 25% of patients might experience vomiting after spinal opioid administration. Nausea often precedes vomiting, but they can also occur separately. Many patients receiving opioids rate the nausea and vomiting as worse than their pain [30]. Recent studies emphasized that postoperative nausea and vomiting were triggered by hypotension most of the time [12,31,32]. During spinal anesthesia for cesarean section, these problems can be exacerbated by uterine manipulation and peritoneal closure [33]. Some authors advocated that adding an opioid to a local anesthetic in spinal anesthesia would decrease the requirement for intra-operative antiemetic drugs, effects provided mostly by intrathecal fentanyl and not by sufentanyl [34,35]. In a recent systematic review completed with a Bayesian network meta-analysis conducted by Hiroyuki Seki and colleagues [12], clinically relevant opioid-related adverse effects were investigated, but the results concerning the vomiting were inconsistent. They concluded that, while lipophilic opioids might decrease intraoperative nausea and/or vomiting associated with uterine exteriorization, hydrophilic opioids might exacerbate these side effects [12,36].

One of the most concerning and severe adverse events of spinal administration of opioids is respiratory depression. Most studies reported no respiratory adverse events due to intrathecal opioids. Our study also registered zero events. However, some other research pointed out that sufentayl and morphine were associated with a significant increase in the incidence of respiratory depression. They showed that respiratory depression was associated with spinal administration of morphine at doses higher than 1.0 mg [16].

In our study, dizziness on first mobilization after spinal anesthesia was often encountered in the morphine group (55.3% compared to 24.1% in the fentanyl group). This dizziness reported to be associated with mobilization was transient and short-lived, with no clinical consequence and no distress for the patient. Constant dizziness was reported in only three cases (1.7%), all associated with morphine administration. Other studies reported dizziness or strange feelings occurring shortly after the spinal anesthesia and associated with hypotension [30].

Although a study on the use of morphine and fentanyl for cesarean section might seem redundant, it is of particular interest for the female population in Eastern Europe, where the number of cesarean sections is considerably high and the reasons that underlie the decision are very diverse. Eurostat data on the percentage of cesarean births in EU countries in 2017 revealed the highest rates in Cyprus (54.8% of all live births being performed via cesarean section), followed by Romania (44.1%), and Bulgaria (43.1%) [37]. A recent paper by Petre et al. reported even higher rates in Romania, ranging from 53.6% to 60.7% [38]. Various factors, both personal and external, contribute to this excessive use of cesarean sections. At the individual level, factors like the fear of pain, cultural beliefs, and the desire for increased safety for both mother and child influence a woman’s decision [39,40,41]. It is worth noting that a substantial number of cesarean operations can also create financial strain on the national health system. Better pain management for parturients could reduce hospital stay and the resulting financial burden.

## 5. Limitations

We acknowledge that per-protocol analysis entails a risk of attrition bias. Even further, the outcome is influenced by factors related to adherence to treatment, and we faced different dropout rates in the two groups (one versus sixteen in the M and F groups, respectively). To minimize the number of patients excluded from the final analysis, imputation was applied in cases of emergency pain therapy; i.e., the last observation was carried forward; this approach might have led to underestimated variability in the pain scores in F group, in which such rescue medication was required. An additional limitation concerns the dosage investigated: one standard dose was used for each of the two opioids considered in the present study.

## 6. Conclusions

According to our findings, morphine for spinal anesthesia in cesarean section offers better postoperative analgesia compared to fentanyl. On the other hand, morphine was associated with a higher chance of non-life-threatening side effects, such as pruritus, nausea, and dizziness. Despite these analgesic-related adverse effects, patients who received morphine reported higher effectiveness of analgesia and better overall satisfaction. Future clinical trials should be conducted to evaluate the adequate doses of opioids and local anesthetics for optimal spinal anesthesia with safe side effects profiles.

## Figures and Tables

**Figure 1 jcm-12-06346-f001:**
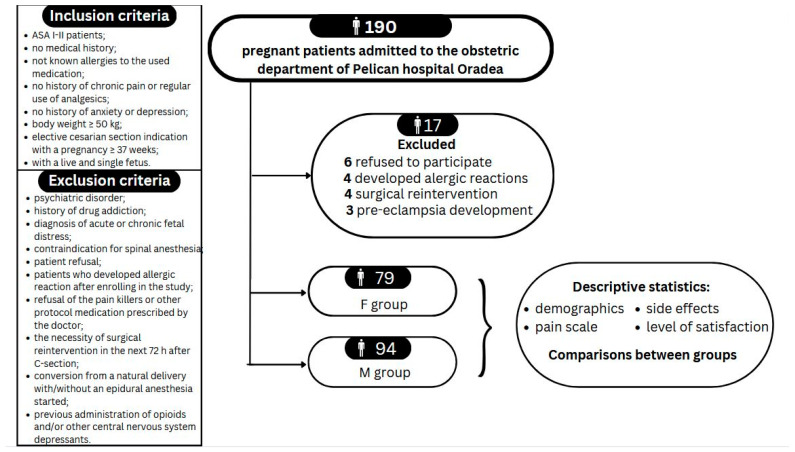
Study design.

**Figure 2 jcm-12-06346-f002:**
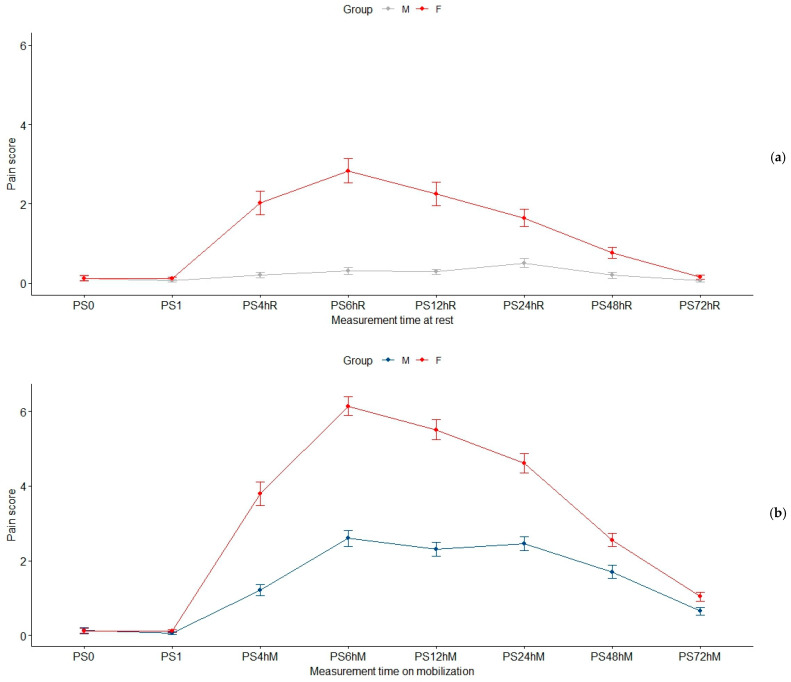
Pain scores in time for the two groups: (**a**) motionless or at rest, and (**b**) on mobilization. At each moment, the mean score and standard error are depicted. The reported scores for intra-operative (PS0) and immediate postoperative (PS1) pain are shown as references in both diagrams.

**Figure 3 jcm-12-06346-f003:**
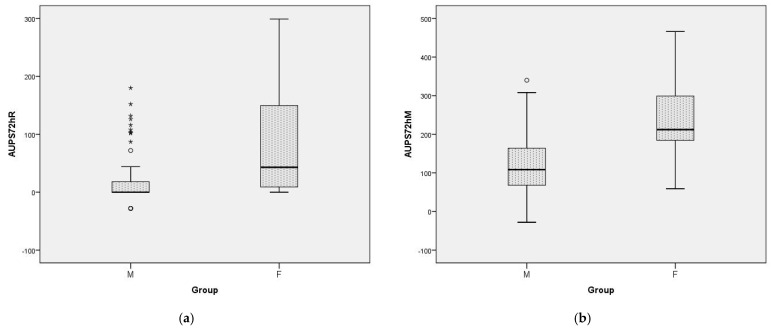
Area under pain intensity scores over 72 h (AUPS72h) for the two groups: (**a**) motionless or at rest, and (**b**) on mobilization. Boxes are proportional to the interquartile range (IQR) with medians marked in between, and the whiskers are proportional to 1.5*IQR (or trimmed to the minimum or maximum values). The bullets and stars are outliers and extreme values, respectively.

**Figure 4 jcm-12-06346-f004:**
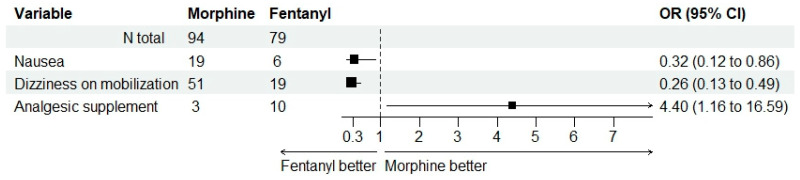
Odds ratio values (with their corresponding 95% confidence intervals) for the symptoms associated with anesthesia on the one hand and need for additional analgesics on the other hand. The very large confidence interval for the need of supplements can be observed, thus inferring more uncertainty related to this possible outcome, in contrast to the favorable odds regarding possible symptoms.

**Table 1 jcm-12-06346-t001:** Descriptive statistics of preoperative data.

Characteristic/ Variable	All Patients (N = 173)	Morphine Group (N = 94)	Fentanyl group (N = 79)	*p*-Value ^(a),(b)^
Age in years ^(a)^	31.88 ± 4.64	31.93 ± 4.43	31.84 ± 4.92	0.899
Weight in kg ^(a)^	76.69 ± 13.43	78.00 ± 14.00	75.14 ± 11.18	0.153
Height in cm ^(a)^	164.79 ± 5.72	165.00 ± 6.20	164.53 ± 5.11	0.587
Smoker ^(b)^	24 (13.9%)	17 (18.1%)	7 (8.9%)	0.039 *
Gesta ^(b)^				0.245
0	93 (53.8%)	50 (53.2%)	43 (54.4%)	
1	59 (34.1%)	32 (34%)	27 (34.2%)	
2	12 (6.9%)	4 (4.3%)	8 (10.1%)	
3	9 (5.2%)	8 (8.5%)	1 (1.3%)	
Para ^(b)^				0.409
0	104 (60.1%)	56 (59.6%)	48 (60.8%)	
1	58 (33.5%)	32 (34%)	26 (32.9%)	
2	9 (5.2%)	4 (4.3%)	5 (6.3%)	
3	2 (1.2%)	2 (2.1%)	–	
Preoperative contractions ^(b)^	33 (19.1%)	17 (18.1%)	16 (20.3%)	0.144

^(a)^ mean ± standard deviation; normal distribution; *t*-test for independent samples. ^(b)^ observed frequency (percentage); chi-squared test (either asymptotic, Fisher’s exact test, or Monte-Carlo simulation with 10,000 samples, as appropriate). Statistical significance: * *p* < 0.05.

**Table 2 jcm-12-06346-t002:** Time elapsed from anesthesia to sensory block and surgical incision.

Characteristic/ Variable	All Patients (N = 173)	Morphine Group (N = 94)	Fentanyl Group (N = 79)	*p*-Value ^(a)^
Time to sensory block, in minutes ^(a)^	3.58 ± 0.96	3.52 ± 1.03	3.65 ± 0.86	0.234
Time from anesthesia to incision, in minutes ^(a)^	6.50 ± 1.69	6.71 ± 1.82	6.25 ± 1.49	0.165

^(a)^ mean ± standard deviation; non-normal distribution; Mann–Whitney U test.

**Table 3 jcm-12-06346-t003:** Perioperative pain (pain intensity was quantified on a scale from 0 to 10).

Characteristic/ Variable	All Patients (N = 173)	Morphine Group (N = 94)	Fentanyl Group (N = 79)	*p*-Value ^(a)^
AUPS72hR^(a)^	47.07 ± 73.35	17.95 ± 39.47	81.72 ± 88.14	<0.001 **
AUPS72Hm ^(a)^	174.00 ± 103.53	124.35 ± 81.23	233.08 ± 96.25	<0.001 **
PS0 ^(a)^	0.13 ± 0.64	0.13 ± 0.74	0.13 ± 0.52	0.234
PS1 ^(a)^	0.09 ± 0.36	0.06 ± 0.35	0.11 ± 0.36	0.075
PS4hR ^(a)^	1.04 ± 2.05	0.21 ± 0.65	2.03 ± 2.63	<0.001 **
PS4hM ^(a)^	2.39 ± 2.54	1.21 ± 1.48	3.80 ± 2.81	<0.001 **
PS6hR ^(a)^	1.46 ± 2.28	0.31 ± 0.83	2.84 ± 7.11	<0.001 **
PS6hM ^(a)^	4.21 ± 2.78	2.60 ± 2.01	6.14 ± 2.30	<0.001 **
PS12hR ^(a)^	0.88 ± 2.05	0.29 ± 0.67	2.25 ± 2.57	<0.001 **
PS12hM ^(a)^	3.77 ± 2.65	2.31 ± 1.84	5.51 ± 2.42	<0.001 **
PS24hR ^(a)^	1.03 ± 1.67	0.51 ± 1.15	1.65 ± 1.96	<0.001 **
PS24hM ^(a)^	3.44 ± 2.29	2.46 ± 1.77	4.61 ± 2.30	<0.001 **
PS48hR ^(a)^	0.46 ± 1.05	0.20 ± 0.76	0.76 ± 1.25	<0.001 **
PS48hM ^(a)^	2.09 ± 1.68	1.70 ± 1.68	2.56 ± 1.56	<0.001 **
PS72hR ^(a)^	0.10 ± 0.43	0.06 ± 0.38	0.15 ± 0.48	0.068
PS72hM ^(a)^	0.83 ± 1.05	0.65 ± 0.97	1.04 ± 1.12	0.007 **

^(a)^ mean ± standard deviation; non-normal distribution; Mann–Whitney U test; abbreviations: AUPS72h, area under pain intensity scores over 72 h; M, mobilization; PS, pain score; R, at rest; statistical significance: ** *p* < 0.01.

**Table 4 jcm-12-06346-t004:** Adverse effects associated with anesthesia.

Characteristic/ Variable	All Patients (N = 173)	Morphine Group (N = 94)	Fentanyl Group (N = 79)	*p*-Value ^(a),(b)^
Pruritus on a 0–4 scale ^(a)^	0.68 ± 0.68	0.90 ± 0.73	0.41 ± 0.49	<0.001 **
Pruritus moderate to severe ^(b)^	14 (8.1%)	14 (14.9%)	–	<0.001 **
Nausea ^(b)^	25 (14.5%)	19 (20.2%)	6 (7.6%)	0.019 *
	OR = 0.324; 95%CI (0.123;0.858)			
Vomiting ^(b)^	12 (6.9%)	12 (12.8%)	–	0.001
Sedation ^(b)^	–	–	–	–
Respiratory depression ^(b)^	–	–	–	–
Dizziness on mobilization ^(b)^	71 (41%)	51 (55.3%)	19 (24.1%)	<0.001 **
	OR = 0.256; 95%CI (0.133;0.493)			
Constant dizziness ^(b)^	3 (1.7%)	3 (3.2%)	–	0.251

^(a)^ mean ± standard deviation; non-normal distribution; Mann–Whitney U test; ^(b)^ observed frequency (percentage); chi-squared test (either asymptotic, Fisher’s exact test, or Monte-Carlo simulation with 10,000 samples, as appropriate); abbreviation: OR, odds ratio; statistical significance: * *p* < 0.05; ** *p* < 0.01.

**Table 5 jcm-12-06346-t005:** Need for postoperative rescue analgesia.

Characteristic/ Variable	All Patients (N = 173)	Morphine Group (N = 94)	Fentanyl Group (N = 79)	*p*-Value ^(a)^
Rescue medication ^(a)^	13 (7.5%)	3 (3.2%)	10 (12.7%)	0.019 *
	OR = 4.396; 95%CI (1.165;16.582)			

^(a)^ observed frequency (percentage); chi-squared test (asymptotic); abbreviation: OR, odds ratio; statistical significance: * *p* < 0.05.

**Table 6 jcm-12-06346-t006:** Patients’ feedback: analgesia effectiveness and subjective satisfaction.

Characteristic/ Variable	All Patients (N = 173)	Morphine Group (N = 94)	Fentanyl Group (N = 79)	*p*-Value ^(a),(b),(c)^
Analgesia was effective ^(a)^	155 (89.9)	94 (100%)	61 (77.2%)	<0.001 **
Satisfaction on a 1–5 Likert-type scale ^(b),(c)^	4.75 ± 0.58	5 (constant)	4.46 ± 0.77	<0.001 **
	5 (5 – 5)	5 (constant)	5 (4 – 5)	

^(a)^ observed frequency (percentage); chi-squared test (either asymptotic, Fisher’s exact test, or Monte-Carlo simulation with 10,000 samples, as appropriate). ^(b)^ mean ± standard deviation; non-normal distribution; Mann–Whitney U test; ^(c)^ median (interquartile range) with Tukey’s hinges; statistical significance: ** *p* < 0.01.

## Data Availability

Raw data will be made available by the first author, without undue reservation.

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
