# Peer review of "Perioperative Analgesia and Patients’ Satisfaction in Spinal Anesthesia for Cesarean Section: Fentanyl Versus Morphine"

_jcm, 2023, doi:10.3390/jcm12196346_

Round 1
Reviewer 1 Report
Perioperative Analgesia and Patients’ Satisfaction in Spinal Anesthesia for Cesarean Section: Fentanyl versus Morphine
Thank you very much for letting me review a manuscript that addresses a topic of interest and that, in turn, is so unknown. Next, I indicate some very important considerations that must be considered in order to be published.
• Abstract: Indicate the meaning of the acronyms F and M on lines 26 and 27. It is not clear what the conclusions are.
• Introduction: The objectives indicated in the introduction do not correspond to the one that appears in the summary. Please unify.
• Methodology: The content of point 2.3 should come before that of point 2.2. Line 125 should include the characteristics of these variables (dichotomous, continuous, discrete, etc., as well as the different categories of each of them). It is not clear how the pain assessment has been carried out, please clarify.
The analyzes should have been carried out separately in men and women since neither the pain threshold nor the metabolization of the drugs is the same. This may imply a bias in the results obtained.
• Results:
In Table 1, indicate the size of the effect or the confidence intervals of the %.
Please indicate results separately for women and men.
Reviewer 2 Report
This is a well written manuscript with sound results and well-described study design. The major flaw is that it is already well known what the relative benefits of both intrathecal fentanyl and intrathecal morphine provide for patients undergoing cesarean section. It is unclear what value this study provides. It is pretty standard for both fentanyl and morphine to be added to bupivacaine for intrathecal dosing of spinal anesthetic block for this purpose. What do the authors believe this adds to the literature and/or how this would impact or change practice?
Author Response
Thank you for your observations! According to your suggestions, we tried to better explain the added value of our manuscript and we made the following correction:
We added a paragraph to the discussion chapter that we hope will better explain the reason for choosing to approach pain in this particular female population from Eastern Europe. ”Although a study on the use of morphine and fentanyl for cesarean section might seem redundant, it is of particular interest for the female population in the Eastern Europe where the number of cesarean sections is considerably high, and the reasons which underlie the decision are very diverse. Eurostat data on the percentage of cesarean births in EU countries in 2017 revealed the highest rates in Cyprus (54.8 % of all live births being performed via cesarean section), followed by Romania (44.1 %), and Bulgaria (43.1 %) [37]. A recent paper by Petre et al. reported even higher rates in Romania, ranging from 53.6% to 60.7% [38]. Various factors, both personal and external, contribute to this excessive use of Cesarean sections. At the individual level, factors like the fear of pain, cultural beliefs, and the desire for increased safety for both mother and child influence a woman's decision [39–41]. It is worth noting that a substantial number of cesarean operations can also create financial strain on the national health system. Better pain management for parturients could reduce hospital stay and the resulting financial burden.”
Round 2
Reviewer 2 Report
Thank you for your revisions, it is improved.